# Comment on "Origin of water in the Badain Jaran Desert, China: new insight from isotopes" by Wu et al. (2017)

**Lucheng Zhan[1], Jiansheng Chen[2], Ling Li[3] and D. A. Barry[4]**

[1] State Key Laboratory of Hydrology-Water Resources and Hydraulic Engineering, Hohai University, Nanjing, 210098, China

[2] College of Earth Sciences and Engineering, Hohai University, Nanjing, 210098, China

[3] School of Civil Engineering, the University of Queensland, St. Lucia, QLD 4072, Australia

[4] Laboratoire de technologie écologique (ECOL), Institut d'ingénierie de l'environnement (IIE), Faculté de l'environnement naturel, architectural et construit (ENAC), Ecole Polytechnique Fédérale de Lausanne (EPFL), Station 2, 1015 Lausanne, Switzerland

*Correspondence to*: Jiansheng Chen (jschen@hhu.edu.cn)

**Abstract**

Precipitation isotope data were used to determine the origin of groundwater in the Badain Jaran
Desert (BJD) in the study of Wu et al. (2017). Both precipitation and its isotope composition
vary seasonally, so arithmetic averages of precipitation isotope values poorly represent the
isotope composition of meteoric water. Their finding that the BJD groundwater is recharged by
modern meteoric water from local areas including the southeastern adjacent mountains was
based on arithmetic averaging. However, this conclusion is not supported by the corrected mean
precipitation isotope values, which are weighted by the precipitation rate. Indeed, the available
isotopic evidence shows that modern precipitation on the Qilian Mountains is more likely to be
the main source of the groundwater and lake water in the BJD, as found by Chen et al. (2004).
**1 Introduction**
The Badain Jaran Desert (BJD) is characterized by a unique landscape that contains a large
number of lakes and the world's largest stationary sand dunes maintained by groundwater (Chen
et al., 2004). However, the origin of the groundwater remains uncertain (Dong et al., 2013).
Using stable and radioactive environmental isotopes, Wu et al. (2017) investigated the
connection between lakes and groundwater, and the origin of groundwater in the southeastern
desert area. They suggested that the BJD groundwater is derived primarily from modern
meteoric water from local areas, including the southeastern adjacent small mountains. Based
on isotopic evidence, the authors ruled out other hypotheses on the groundwater source,
including fossil groundwater (Gates et al., 2008; Ma and Edmunds, 2006; Wang et al., 2015;
Yang et al., 2010) and snowmelt from the Qilian Mountains, 500 km (center-to-center distance)
southwest of the desert (Chen et al., 2004; 2006).
The authors argued that the $^{14}C$ dating over-estimated the age (~10 ka) of the BJD
groundwater due to interference by additional dead carbon input from ancient carbonates. We
have conducted work related to the $^{14}C$ dating and found the same problem with overestimation
of the groundwater age (Chen et al., 2014; Wang and Chen, 2018). Wu et al. (2017) reasoned
that the average age of groundwater in the BJD should be much younger, since it includes
modern meteoric water as indicated by tritium levels (Gates et al., 2008; Wu et al., 2017). They
presented many evidences and discussions for their conclusion of groundwater recharged by
modern precipitation from local areas. However, their averaging of the precipitation isotope
data did not account for seasonality of precipitation amount, which led to a misconception of
the potential groundwater origin.
**2 Source water identification based on weighted mean precipitation isotope values**
The determination of mean precipitation isotope values is of great significance for assessing the
contribution of precipitation as a water source to regional hydrological systems and for
assessing interactions among different hydrological components. To examine the relationship
between the BJD groundwater and local precipitation, historical precipitation isotope data from
a nearby GNIP (Global Network of Isotopes in Precipitation,
https://nucleus.iaea.org/wiser/index.aspx) station in Zhangye (1986–2003) were used by Wu et
al. (2017). The GNIP database provides data on monthly precipitation isotopes as well as
monthly rainfall for the Zhangye station. As shown in Figure 1a, b, the monthly δD and δ$^{18}$O
values in the study area exhibit large seasonal variations, which are mainly controlled by
temperature (Zhan et al., 2017). The isotopic seasonality pattern of precipitation is similar in
different years. During the summer half year when temperature is higher, the rainfall is more
enriched in heavier isotopes.
According to the GNIP data, the mean annual precipitation is about 130 mm, with more
than 60% of the total annual rainfall occurring from June to August during which the isotope
values are the highest (Figure 1b). Since the annual precipitation is seasonal, the monthly
precipitation isotope data should be weighted by the monthly precipitation amount to calculate
the annual mean for representing the isotope composition of local precipitation as a potential
source of the BJD groundwater. The weighted mean isotopic values $\overline{\delta_w}$ can be calculated
using:
$$\overline{\delta_w} = \frac{\sum\limits_{j=Jan}^{Dec} \overline{\delta_j} \cdot \overline{P_j}}{\sum\limits_{j=Jan}^{Dec} \overline{P_j}} \tag{1}$$

where $\overline{\delta_j}$ and $\overline{P_j}$ are the averaged isotopic values and averaged rainfall amount of month j

during the GNIP observation years, respectively. The monthly averages, $\overline{\delta_j}$ and $\overline{P_j}$ can be

calculated as follows:

$$\overline{\delta_j} = \frac{\sum_i \delta_{i,j}}{n} \tag{2}$$

$$\overline{P_j} = \frac{\sum_i P_{i,j}}{n} \tag{3}$$

where $\delta_{i,j}$ and $P_{i,j}$ are the isotopic value ($\delta$D or $\delta^{18}$O) and rainfall amount of month j in

year i from the available GNIP dataset, respectively; and n is the corresponding number of

available data.

Based on the dataset from the GNIP database, the calculated weighted mean values for $\delta$D

and $\delta^{18}$O of Zhangye's precipitation are -40.9‰ and -5.50‰, respectively (Figure 1c). Using

arithmetic averages, Wu et al. (2017) determined $\delta$D and $\delta^{18}$O values around -74‰ and -10.5‰,

respectively. When plotted on the $\delta^{18}$O-$\delta$D graph (Figure 1c), the arithmetic average values are

close to the intersection of the evaporation line EL1 (for groundwater and lake water in the

desert) and the GMWL (Global Meteoric Water Line), which led Wu et al. (2017) to conclude

that groundwater and lake water in the BJD originates from modern meteoric precipitation in

local areas including the adjacent small mountains. However, if the weighted mean values are

used, this conclusion no longer holds. Compared with the isotope composition of the local

precipitation, the source water recharging the BJD groundwater and lakes is much more

depleted in D and $^{18}$O.

**3 Reanalysis on the origin of groundwater in the BJD**

Using available data from literature, we reanalyzed the possible origin of groundwater in the

BJD. We focus on the BJD southern margin area where the desert lakes are mostly concentrated.

The isotope data of the groundwater and lake water (Figure 2a) lie on the evaporation line EL2

($\delta$D = 4.6$\delta^{18}$O – 29.8, $r^2$ = 0.94), which is reasonably similar to EL1 in Wu et al. (2017). Here

only data from groundwater and lake water samples within the BJD area were used for
determining the EL2. The weighted mean isotope values of precipitation in the regions close to
the BJD (blue circles) show a decreasing trend with increasing elevation from 1382 to 2569 m
a.s.l., reflecting the effect of elevation on isotope fractionation (Poage and Chamberlain, 2001).
The intersection of EL2 and GMWL ($\delta D = -83.6‰$, $\delta^{18}O = -11.7‰$), which represents the mean
isotope composition of the recharge source for BJD groundwater, is outside the range of
precipitation in the local and adjacent regions, indicating another different source water with
more depleted isotope composition.

Together with the statistical isotopic values of precipitation in the BJD and the Qilian

Mountains (rainfall and snowmelt) from literature data, a significant inverse correlation of $\delta D$
and $\delta^{18}O$ values with elevation of the precipitation can be established (Figure 2b, c). The altitude
gradients for $\delta D$ and $\delta^{18}O$ are -2.0‰/100m and -0.26‰/100m, respectively, which are close to
global mean levels (Poage and Chamberlain, 2001). Based on these gradients, the location of
water associated with the intersection of EL2 and GMWL corresponds to an average elevation
of 3914 m a.s.l. (3920 m estimated by $\delta D$ and 3908 m by $\delta^{18}O$). Therefore, the recharge region
for groundwater and lake water in the BJD is likely to include areas of elevations higher than
3914 m a.s.l. to produce source water of more depleted isotope composition.

The closest region that could meet this elevation requirement is the Qilian Mountains

(average elevation between 4000 and 5000 m a.s.l.), northeast of the Qinghai-Tibet Plateau
(Figure 3a). Nineteen snowmelt and rainfall water samples from 3540 to 5010 m a.s.l. in the
glacier zone of the Qilian Mountains were collected by Ren (1999). The statistical isotope
compositions of these samples are close to that given by the GMWL-EL2 intersection (Figure
2a). Therefore, the isotope evidence points to the Qilian Mountains as a main source region for
groundwater and lake water in the BJD, as observed previously (Chen et al., 2004).

Wu et al. (2017) ruled out the Qilian Mountains as a recharge area for groundwater in the

BJD based on the large isotopic difference between the GMWL-EL2 intersection and data from
water samples mainly collected from the Shiyang River watershed (Li et al., 2016), which is
located in the eastern lower area of the Qilian Mountains. The mean elevation of the Shiyang
River watershed is only 2487 m a.s.l. (Bourque and Hassan, 2009), which is lower than the
mean elevation of the entire mountain. Therefore, their argument for excluding the Qilian
Mountains as a recharge region is questionable. Water samples collected from rivers on the
northern slope of the Qilian Mountains are characterized by large variations of isotope
compositions (Figure 2a), with the lowest isotopic values found by Ren (1999) from a river in
the upstream glacier zone. Scattered data between the plots of snowmelt on the mountain and
rainfall in lower regions indicated that most of these river samples are likely to be mixtures of
snowmelt water and piedmont precipitation. Isotope signatures show little connection between
these rivers on the northern slope and the groundwater in the BJD.

The relationship between d-excess and $\delta^{18}O$ was also discussed by Wu et al. (2017). The

d-excess value (d-excess = $\delta D - 8\delta^{18}O < 0$) indicates the deviation from the GMWL, reflecting
the degree of evaporation experienced by the available water. Wu et al. (2017) noted the
difference in the d-excess value between the Qilian-sourced water (sampled from the northern
slope rivers of the Qilian Mountains region) and BJD groundwater, and argued that the Qilian
Mountains cannot be the origin of the latter because no evaporation could occur to water
underground. Located in the northeastern margin of the Qinghai-Tibet Plateau, the Qilian
Mountains area consists of many northwest–southeast parallel mountain ranges and valleys
(Qiu et al., 2016). Although little evidence of evaporation was found in sampled river water
from the northern slope area, water in other near-surface water systems (like lakes, wetlands,
and soil water reservoir) of longer residence time within the wide Qilian Mountains region
would have been subjected to more intense evaporation and produced isotopic signatures
similar to that of the BJD groundwater. The d-excess results cannot exclude the Qilian
Mountains as a recharge region either.

Groundwater in the BJD has also been postulated to be sourced from the Yabulai Mountain

region (Figure 3a). The highest mountain there is 1938 m a.s.l., which is unlikely to provide
rainfall input with depleted heavy isotopes as shown in Figure 2. In a recent groundwater
resource development project, eight wells were drilled (depths of 135 to 260 m) in the
southeastern part of the BJD. The static groundwater levels in these wells show a decreasing
trend from southwest to northeast (Figure 3b), indicating an overall movement of groundwater
along this direction. The groundwater flow direction is consistent with our hypothesis that BJD
groundwater originates from the Qilian Mountains (located southwest of the BJD). Researchers
have also examined the chemistry of lake water and groundwater in the study area and
surrounding areas. For example, Yang and Williams (2003) investigated the ion chemistry of
lake water and groundwater from the BJD and its periphery, and ruled out the possibility of
recharge from recent local rainfall to the lakes and groundwater. In a previous study (Chen et
al., 2012), the hydrochemical and isotopic results also supported our remote recharge
hypothesis.
We agree with the concern of Wu et al. (2017) on the accuracy of $^{14}$C dating for the BJD
groundwater, which provided estimates of very old ages. Recent work (Wang and Chen, 2018)
found considerable overestimation of the groundwater age by the $^{14}$C dating method due to
neglect of dead carbon brought by deep $CO_2$ emission. In contrast to the fossil groundwater
hypothesis, the detectable tritium activities as shown by Wu et al. (2017) and many others (Chen
et al., 2006; Gates et al., 2008; Yang and Williams, 2003) indicate a modern precipitation source
of the BJD groundwater. This suggests that the Qilian Mountains-sourced groundwater flows
through hundreds of kilometers over only tens of years. We suggest that, due to geological
activities, various southwest–northeast deep fault systems exist between the Qilian Mountains
and the desert (Chen et al., 2006). Based on the geological conditions and geochemical
evidences (helium results), these large deep fault systems are hypothesized to act as a quick
passage for the groundwater (Chen et al., 2006, 2004, 2012), which explains the detectable
tritium in the groundwater.
The reanalysis above supports the hypothesis that groundwater in the BJD mainly
originates from modern precipitation of Qilian Mountains. Near-surface water in the Qilian
Mountains, subjected to evaporation, infiltrates and recharges groundwater, which is then
delivered to the BJD through the deep interconnected faults. Of course, more work is needed to
support this hypothesis conclusively. It should also be noted that, the higher average elevation
(4000 to 5000 m a.s.l.) of the Qilian Mountains than the estimated mean recharge elevation
(3914 m a.s.l.) estimated in this study, as well as the large variation of isotope composition of
groundwater in the BJD, may imply a mixture of the Qilian-sourced water (of more depleted
isotope composition from 4000 to 5000 m a.s.l) with precipitation from other lower areas.
Groundwater might have mixed with rainwater from low-elevation areas on its pathway.
**4 Concluding remarks**
We reanalyzed the precipitation isotope data of the Zhangye station to determine the original
source of the groundwater in the Badain Jaran Desert. These data were averaged arithmetically
in the recent study of Wu et al. (2017), whereas weighted averaging is more appropriate. The
reanalysis does not support the conclusion of Wu et al. (2017) that the BJD groundwater is
sourced from local meteoric water. Indeed, the reanalysis suggests a mean recharge elevation
of about 3914 m a.s.l. for the BJD groundwater, which indicates that the precipitation in the
Qilian Mountains region is a more likely main source of the BJD groundwater, as initially
hypothesized by Chen et al. (2004).

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

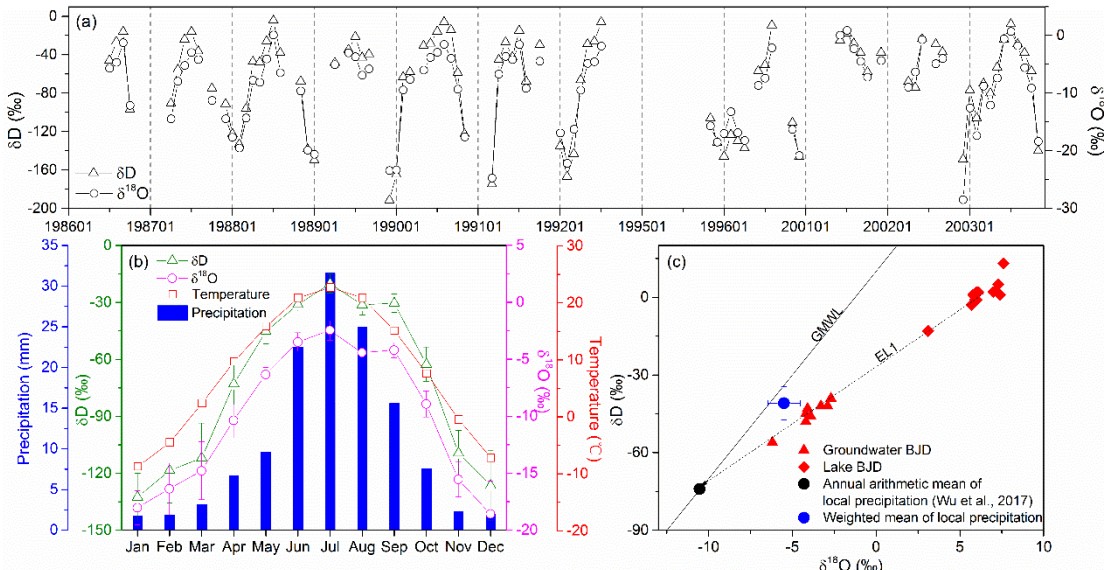

**Figure 1.** Isotope composition of monthly precipitation of the GNIP station Zhangye (all
available datasets **(a)** and monthly mean values **(b)**), and δD-δ$^{18}$O plots of groundwater, lake
water and annual precipitation in the study area (based on data from Zhangye station) **(c)**. Data
in **(a)** and **(b)** are sourced from the GNIP database while plot **(c)** is modified from Wu et al.
(2017). Statistical mean values are shown together with standard errors where feasible.

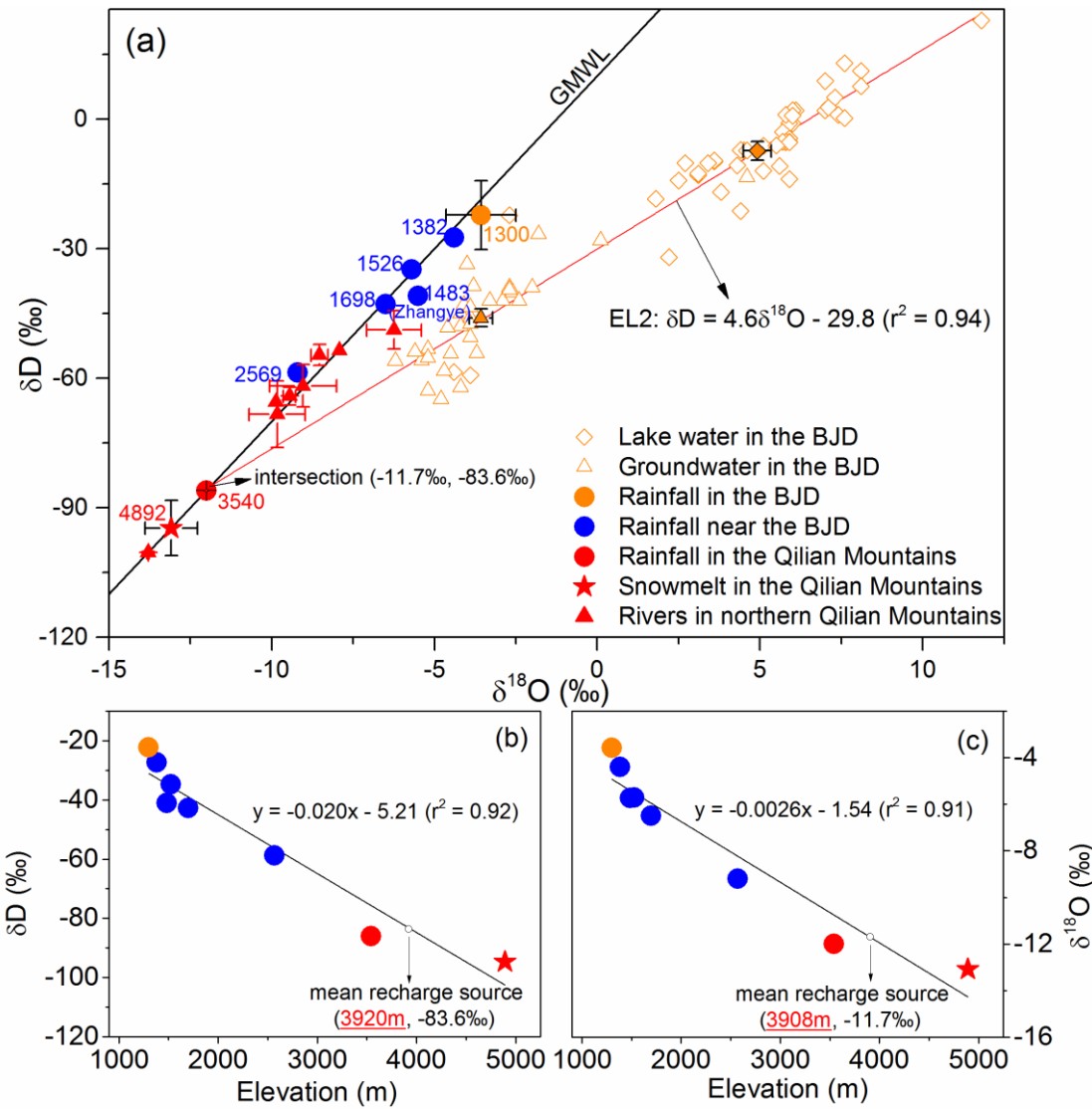

**Figure 2.** δD vs. δ¹⁸O plot of water related to the BJD groundwater origin **(a)**, and altitude gradients of related precipitation isotopes **(b, c)**. For precipitation (rainfall and snowmelt), the corresponding sampling elevations (m a.s.l.) are also shown. Statistical mean values are shown together with standard errors where feasible. The weighted means of local rainfall (blue circles) are from Wu et al. (2010) and the GNIP database. Rainfall (yellow circle), lake water (yellow square; 47 samples) and groundwater (yellow triangle; 31 samples) in within the BJD area are based on data from Wu et al. (2017), Ma and Edmunds (2006), Zhao et al. (2012), Gates et al. (2008), Chen et al. (2012) and Yang et al. (2010). Summer rainfall (red circle; 4 samples) and snowmelt (red pentagram; 15 samples) in the Qilian Mountains are based on data from Ren (1999). Isotopic data for various rivers (red triangles) on the northern slope of the Qilian Mountain are collected from Chen et al. (2012), Li et al. (2016), Zhu, Su, and Feng (2008) and Ren (1999).

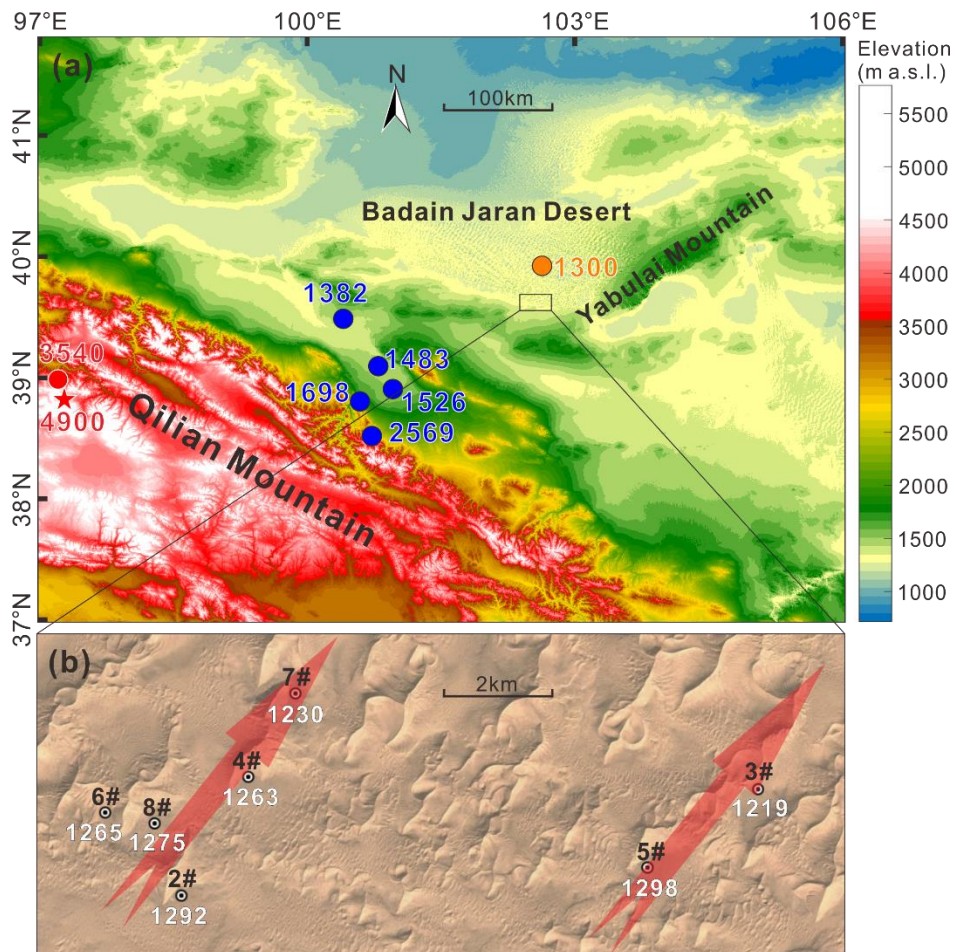

272

**Figure 3.** Elevation map of the Qilian Mountains and BJD areas **(a)** and groundwater wells drilled in the BJD **(b)**. Locations for precipitation sampling in different areas are also shown in **(a)**, as well as the elevation (m a.s.l.). The elevations of static groundwater levels in seven of the extraction wells (well #1 is far away from these wells and hence not shown) are indicated by white text in **(b)**. Arrows in **(b)** show the estimated groundwater flow direction (based on groundwater elevation).