# Peer review of "Comment on "Origin of water in the Badain Jaran Desert, China"

_Hydrology and Earth System Sciences, 2018_

## Referee Comment (RC1) · Anonymous Referee #1 · 11 Jun 2018

The manuscript on "Comment on – "Origin of water in the Badain Jaran Desert, China: new insight from isotopes" by Wu et al. (2017)" - by Lucheng Zhan, et al. provides evidence, that groundwater in the Badain desert is not locally recharged as recommended by Wu et al., (2017) but originating from the Qilian Mountains. This argument is based mainly on a recalculation / corrected values of "an amount weighed stable isotope value of precipitation", which differs significantly from the former value proposed by Wu et al., and additionally on new groundwater observations collected in the area and provided in the paper.

The comment by Zhan et al. is correct and justified. It is arguing against results from

Wu et al. in a sense full and scientifically correct form. The comment is well written and logically sound. In my opinion this is a very nice example how 'stimulating discussions' potentially might push science and knowledge in a right direction. It also highlights pros and cons of isotope based research in a productive way and will eventually lead to a better understanding of the Badain Jaran Desert system for all interested readers.

I recommend to allow and accept the comments for publication in HESS after only minor revisions.

General comments

- A key role in both papers is the data set from the WMO/IAEA on stable isotopes in precipitation – station Zhangye (1986 – 2003, n=86). I recommend, that more information on GNIP station Zhangye and in addition to the mean monthly isotope data set (Fig. 1a) the long-term isotope set is implemented in the work to clarify seasonality and trends of the 17 years data set. - Wu et al. were describing their data point as annual average from 'monthly weighted average' values. Therefore I would recommend that the authors include their weighing formula into the text. Were mean monthly values weighed or monthly values to yearly precipitation? - The authors do not comment on earlier an earlier hypotheses, that groundwater might contribute fossil water (Line 30), which potentially was recharged during cooler periods and therefore with more depleted d2H, d18O values. If this would be the case, elevations would not need to be as high as 3914 m a.s.l. (Line 88).

Minor comments - Line 82: (Figure 2b, c) instead of (Figure 2b&c) - Line 105: Wu et al. 2017 instead of 2016 - Line 144: . . .isotope data of the Zhangye station to determine. . .. - Line 217: . . .monthly precipitation of the GNIP station Zhanye (a). . . . - Line 220: delete: Further details are provided in the text. - Line 222: dD vs. d18O . . . - Line 223: . . . (b, c) . . . instead of (b & c)

Please also note the supplement to this comment:

https://www.hydrol-earth-syst-sci-discuss.net/hess-2018-229/hess-2018-229-RC1-supplement.pdf

---

## Referee Comment (RC2) · Anonymous Referee #2 · 11 Jun 2018

The manuscript 'Comment on "Origin of water in the Badain Jaran Desert, China: new insight from isotopes" by Wu et al. (2017)' contributes to the debate about origin of water discharging in a unique desert ecosystem. The authors refute the main result of the previous HESS paper, drawn from analysis of isotopic data, that local precipitation is the main source of groundwater feeding lakes of the area. The key argument against the observation of Wu et al. is that they incorrectly calculated the annual mean isotopic composition of local precipitation. A reliable quantification of these mean values is therefore a key factor in the evaluation of the manuscript. Unfortunately, neither the authors of the commenting paper, nor Wu et al. present details of their calculations. The GNIP record of isotopic composition of precipitation at Zhangye station covers only 18

years with many gaps in data. Given that, as well as large seasonality of precipitation amount and its isotopic composition for that station, evaluation of the annual means is sensitive to the averaging method which is not described in the manuscript. Putting this issue aside, the reviewed manuscript tries to draw unambiguous conclusions concerning origin of groundwater for a huge system extending over hundreds of kilometers basing them on a limited amount of isotope data. Environmental tracers work best when supported by the understanding of the hydrogeological and hydrogeochemical characteristics of a system, none of which is available in this case. Conclusions based solely on the isotopic composition, especially for the large and diversified system, are subjected to large uncertainties which are completely not assessed here. These uncertainties are enhanced by the pronounced influence of evaporation on the isotopic signatures in such an arid environment. In its present form, the manuscript does not provide a convincing evidence against the results presented by Wu et al., especially that tritium data suggest a significant contribution from recent precipitation. I suggest major revision.

Suggestions for improving the manuscript.

1. Precisely describe methods used to calculate both the weighted and unweighted average isotopic compositions of precipitation.

2. The two disputed components of groundwater – recent infiltration and water recharging in distant mountain chain - should be easily distinguishable by the concentrations or concentration ratios of dissolved components. Are there any data that could be used to identify their chemical signatures?

3. Page 2/line 32. Distance between Qilian Mountains and the desert shown on the map (Fig. 3) seems to be smaller than 500 km.

3. Page 2/lines 35 - 38. The reasoning presented in the last two sentences of page 2 is logically flawed. Incorrect calculation of the averaged isotopic composition of precipitation does not invalidate the meaning and significance of tritium results.

4. Page 5/lines 113 - 117. Are the surface water bodies mentioned here known to recharge groundwater or do hydrogeological conditions allow for infiltration from them?

5. Page 5-6/lines 122 - 127. Recharge in Qilian Mountains cannot be a source of detectable tritium in the desert or we have to assume that groundwater flows over hundreds of kilometers in tens of years. As with point 3, tritium data are not well integrated in the discussion.

Fig. 1. What are standard deviations (due to averaging) of the monthly and annual averages presented here? They should be shown on the plots if significant.

Fig. 2. There is a considerable spread in groundwater isotopic data used to derive EL2 evaporation line, which might lead to a biased identification of the line itself and of its interception with GMWL. These data are pooled results of several studies, do all of them represent locations on the presumed groundwater flow lines between the recharge area and BJD lakes? Perhaps not all of them are representative for derivation of the evaporation line.
* * *

---

## Short Comment (SC1) · 2 Jul 2018

Thanks to the group of Dr. Zhan who wrote this Comment. Now, I find the mistake I have made. So I make the corrigendum below: Due to the ignoring calculation of annually weighted average values, I misused the mean values as the weighted average values of isotope ratios of precipitation at the IAEA-GNIP (IAEA/WMO, 1986-2003) station in Zhangye. So the words "weighted annually average" in the figure caption of Figure 4 in the page of 4225 and "weighted mean" in the section 5.2 (Page 4426, Line 28 left side), which should be "annually average" and "mean", respectively. In addition, I add the true annually weighted average values (IAEA/WMO, 1986-2003) to

the Figure 4 and Figure 5a. As shown in the Figure 4 and Figure 5(a), the weighted average value (black square in the Figure 4&5a) is higher than the intersection of the evaporation line and the GMWL in the $\delta$18O-$\delta$D plot. This may indicate preferential recharge by winter precipitation in such an arid region, unlike in humid region where the groundwater generally has a composition similar to the weighed mean annual precipitation in the watershed. In other words, the local groundwater is perhaps recharged primarily by snowmelt in the area and/or surrounding mountains, with little or no summer rain making it to the water table due to high evaporation and/or the plant evapotranspiration in the summer. I also have two suggestions to this comment: 1. This comment make some good theoretic analysis of the altitude effect on the stable isotopes of precipitation, however, I did not see more isotopic data from the Qilian Mountain. Considering they questioned the representability of the samples from Shiyang River (Li et al., 2016), I think more data from the Qilian Mountain would make this comparison more clearly. 2. In my opinion, the more specific description of the hydrogeological processes and the evolution of water isotopes is necessary to support the remote Qilian Mountain as the major recharge area.

Please also note the supplement to this comment:
https://www.hydrol-earth-syst-sci-discuss.net/hess-2018-229/hess-2018-229-SC1-supplement.pdf

―――――――――――――――

Fig. 1.

- ■ Annually average precipitation  Zhangye
- ● Average precipitation  Zhangye
- ■ Monthly average precipitation  Zhangye
- ◆ Lake  BJD    ▲ Groundwater  BJD
- ✳ Rain  BJD
- —— GMWL
- - - - EL

Summer months

Cold months

$EL_2: y = 4.5x - 27.7$

Y-axis: δD-VSMOW (‰)

X-axis: δ¹⁸O-VSMOW(‰)

EL Yabulai: y=4.2x-24.1

EL $_3$: y=4.4x-30.6

$\delta$D-VSMOW (‰)

Summer months

Cold months

EL $_1$

$\delta^{18}$O-VSMOW(‰)

(c)

GMWL    ■ Monthly weighted average Zhangye
■ Annually weighted average Zhangye  ● Annual average Zhangye
▲ Groundwater BJD          ◆ Lake BJD          ✻ Rain BJD
▼ Deep groundwater BJD margin  ● Shallow groundwater Yabulai  ◻ Xugue
○ Precipitation Qilian   ◇ River Qilian   △ Groundwater Qilian
✶ Glacier snow meltwater  Qilian        ✚ Frozen soil meltwater Qilian

(d)

$d$-excess = -3.4$\delta^{18}$O -30.3

$d$-excess (‰)

▲ Groundwater BJD

◆ Lake BJD

■ Land water Qilian Mountain

● Shallow groundwater Yabulai

◻ Infiltration ES-1 to ES-3

△ Evaporation Pan-1

◇ Evaporation Pan-2

**Fig. 2.**

**Supplement:**

Reply to the Comment on "Origin of water in the Badain Jaran Desert, China: new insight from isotopes" by Wu et al. (2017)

*By Xiujie Wu* (xiujiewu1990@foxmail.com).

Thanks to the group of Dr. Zhan who wrote this Comment. Now, I find the mistake I have made.

So I make the corrigendum below:

5   Due to the ignoring calculation of annually weighted average values, I misused the mean values as the weighted average values of isotope ratios of precipitation at the IAEA-GNIP (IAEA/WMO, 1986-2003) station in Zhangye.

So the words "weighted annually average" in the figure caption of Figure 4 in the page of 4225 and "weighted mean" in the section 5.2 (Page 4426, Line 28 left side), which should be "annually average" and "mean", respectively.

In addition, I add the true annually weighted average values (IAEA/WMO, 1986-2003) to the Figure 4 and Figure 5a.

10   As shown in the Figure 4 and Figure 5(a), the weighted average value (black square in the Figure 4&5a) is higher than the intersection of the evaporation line and the GMWL in the $\delta^{18}$O-$\delta$D plot. This may indicate preferential recharge by winter precipitation in such an arid region, unlike in humid region where the groundwater generally has a composition similar to the weighed mean annual precipitation in the watershed. In other words, the local groundwater is perhaps recharged primarily by snowmelt in the area and/or surrounding mountains, with little or no summer rain making it to the water table due to high

15   evaporation and/or the plant evapotranspiration in the summer.

I also have two suggestions to this comment:

1.   This comment make some good theoretic analysis of the altitude effect on the stable isotopes of precipitation, however, I did not see more isotopic data from the Qilian Mountain. Considering they questioned the representability of the samples from Shiyang River (Li et al., 2016), I think more data from the Qilian Mountain would make this comparison more

20      clearly.

2.   In my opinion, the more specific description of the hydrogeological processes and the evolution of water isotopes is necessary to support the remote Qilian Mountain as the major recharge area.

[Figure]

**Figure 4: The δD vs. δ¹⁸O plot of natural groundwater, lake water, and precipitation in the desert. Also shown are weighted monthly average and weighted annually average isotope ratios of precipitation at the IAEA-GNIP station in Zhangye.**

[Figure]

**Figure 5: The plot of δD vs δ¹⁸O values (a) and *d*-excess vs δ¹⁸O values (b) of groundwater and lake water samples from the BJD (red symbols), including new data from his study and previously published data from the literature (Gates et al., 2008a; Zhang et al., 2011; Zhao et al., 2012). The trend line in (b) is established from our evaporation experiments (Fig.4b). Also shown are the isotope data from the Qilian Mountain area (light blue symbols) for comparison. The two larger diamond dots with black cross inside are the average values with error bar for the Sumu Jaran and Sumu Badain Jaran lakes sampled at different depths. Isotope data for deep groundwater in Gurinai and Xugue and shallow groundwater in Xugue and Yabulai Mountains in (A) are from Gates et al. (2008a). The water isotope data for the Qilian Mountains include precipitation (Wu et al., 2010; Chen et al., 2012), and land water including groundwater (Li et al., 2016), rivers (average for each river) (Chen et al., 2012; Li et al., 2016), glacier snow melt water and frozen soil melt water (Li et al., 2016).**

**References**

IAEA/WMO, 1986-2003. Global Network of Isotopes in Precipitation. <http://www.iaea.org/water, 2018>.

Li, Z.X., Feng, Q., Wang, Q. J., Yong, S., Cheng, A. F., and Li, J. G.: Contribution from frozen soil meltwater to runoff in an in-land river basin under water scarcity by isotopic tracing in northwestern China, Global Planet. Change., 136, 41-51, doi: 10.1016/j.gloplacha.2015.12.002, 2016.

---

## Author Comment (AC1) · 13 Jul 2018

**Response to anonymous referee #1**

The manuscript on "*Comment on – "Origin of water in the Badain Jaran Desert, China: new insight from isotopes" by Wu et al. (2017)*" - by Lucheng Zhan, et al. provides evidence, that groundwater in the Badain desert is not locally recharged as recommended by Wu et al., (2017) but originating from the Qilian Mountains. This argument is based mainly on a recalculation / corrected values of "an amount weighed stable isotope value of precipitation", which differs significantly from the former value proposed by Wu et al., and additionally on new groundwater observations collected in the area and provided in the paper.

The comment by Zhan et al. is correct and justified. It is arguing against results from Wu et al. in a sense full and scientifically correct form. The comment is well written and logically sound. In my opinion this is a very nice example how 'stimulating discussions' potentially might push science and knowledge in a right direction. It also highlights pros and cons of isotope based research in a productive way and will eventually lead to a better understanding of the Badain Jaran Desert system for all interested readers.

I recommend to allow and accept the comments for publication in HESS after only minor revisions.

**Response:** Thanks for your positive comments. We will further improve the paper following the suggestions of both referees. The detailed responses to the specific comments and corresponding revisions we plan to make are listed below.

**General comments**

- A key role in both papers is the data set from the WMO/IAEA on stable isotopes in precipitation – station Zhangye (1986 – 2003, n=86). I recommend, that more information on GNIP station Zhangye and in addition to the mean monthly isotope data set (Fig. 1a) the long-term isotope set is implemented in the work to clarify seasonality and trends of the 17 years data set.

**Response and changes to the manuscript:** We agree with your suggestion. Precipitation isotope composition is the key point of this comment paper and more detailed analysis on its changing pattern is needed. In our revision, Figure 1 will be modified to show more information about the seasonality and trends of precipitation isotope in the study area, with related discussions added in the text.

- Wu et al. were describing their data point as annual average from 'monthly weighted average' values. Therefore I would recommend that the authors include their weighing formula into the text. Were mean monthly values

weighed or monthly values to yearly precipitation?

**Response and changes to the manuscript:** Another referee also suggested addition of detailed descriptions of the methods used to calculate both the weighted and unweighted average isotopic compositions of precipitation. We agree with your concern and will include these descriptions in the text.

- The authors do not comment on earlier an earlier hypotheses, that groundwater might contribute fossil water (Line 30), which potentially was recharged during cooler periods and therefore with more depleted d2H, d18O values. If this would be the case, elevations would not need to be as high as 3914 m a.s.l. (Line 88).

**Response and changes to the manuscript:** In terms of the fossil groundwater hypothesis, our opinion is consistent with that of Wu et al. (2017). As discussed in the text (lines 119-127), the fossil water opinion is questionable because of the overestimation of groundwater age by the $^{14}$C dating and detectable tritium in the groundwater. For clarity, more analysis and discussion on the tritium data of groundwater and its residence time will be added in the revised manuscript to further support our hypothesis.

**Minor comments**

- Line 82: (Figure 2b, c) instead of (Figure 2b&c)

**Response and changes to the manuscript:** This will be revised as you suggested.

- Line 105: Wu et al. 2017 instead of 2016

**Response and changes to the manuscript:** This mistake will be corrected.

- Line 144: …isotope data of the Zhangye station to determine….

**Response and changes to the manuscript:** Agree. This sentence will be revised.

- Line 217: …monthly precipitation of the GNIP station Zhanye (a). …

**Response and changes to the manuscript:** This sentence will be revised following your suggestion.

- Line 220: delete: Further details are provided in the text.

**Response and changes to the manuscript:** This sentence will be deleted.

- Line 222: dD vs. d18O …

**Response and changes to the manuscript:** This sentence will be changed to "δD vs. δ$^{18}$O plot of water related to…"

- Line 223: … (b, c) … instead of (b & c)

**Response and changes to the manuscript:** It will also be revised.

**Response to anonymous referee #2**

The manuscript 'Comment on "Origin of water in the Badain Jaran Desert, China: new insight from isotopes" by Wu et al. (2017)' contributes to the debate about origin of water discharging in a unique desert ecosystem. The authors refute the main result of the previous HESS paper, drawn from analysis of isotopic data, that local precipitation is the main source of groundwater feeding lakes of the area. The key argument against the observation of Wu et al. is that they incorrectly calculated the annual mean isotopic composition of local precipitation. A reliable quantification of these mean values is therefore a key factor in the evaluation of the manuscript. Unfortunately, neither the authors of the commenting paper, nor Wu et al. present details of their calculations. The GNIP record of isotopic composition of precipitation at Zhangye station covers only 18 years with many gaps in data. Given that, as well as large seasonality of precipitation amount and its isotopic composition for that station, evaluation of the annual means is sensitive to the averaging method which is not described in the manuscript. Putting this issue aside, the reviewed manuscript tries to draw unambiguous conclusions concerning origin of groundwater for a huge system extending over hundreds of kilometers basing them on a limited amount of isotope data. Environmental tracers work best when supported by the understanding of the hydrogeological and hydrogeochemical characteristics of a system, none of which is available in this case. Conclusions based solely on the isotopic composition, especially for the large and diversified system, are subjected to large uncertainties which are completely not assessed here. These uncertainties are enhanced by the pronounced influence of evaporation on the isotopic signatures in such an arid environment. In its present form, the manuscript does not provide a convincing evidence against the results presented by Wu et al., especially that tritium data suggest a significant contribution from recent precipitation. I suggest major revision.

**Response:** Thanks for your valuable comments. We agree with your concerns about the uncertainties of this study due to limited isotope data. As a comment paper, it mainly focused on the results of the original paper. We aimed to correct the incorrect usage of precipitation isotope data in the original paper and show that the published results can be interpreted differently based on a more appropriate analysis of the data. We tried to collect more available data and present our own recent results to support our hypothesis. Nonetheless, we recognise the need to further improve this manuscript to make it more convincing. Following your suggestions, discussions on groundwater chemical signatures will be added to further support our hypothesis. In addition, more analysis of the tritium data in relation to the possible groundwater recharging mechanism will be further examined. The detailed responses to your specific suggestions are listed below.

Suggestions for improving the manuscript.

1. Precisely describe methods used to calculate both the weighted and unweighted average isotopic compositions of precipitation.

**Response and changes to the manuscript:** Agree. More details about the calculations of the weighted and arithmetic averages of precipitation isotope, including the specific dataset and calculating formulas, will be added in the text.

2. The two disputed components of groundwater – recent infiltration and water recharging in distant mountain chain - should be easily distinguishable by the concentrations or concentration ratios of dissolved components. Are there any data that could be used to identify their chemical signatures?

**Response and changes to the manuscript:** We agree with your opinion. Some researchers have also investigated the chemistry of lake water and groundwater in the study area and surrounding areas. For example, Yang and Williams (2003) investigated the ion chemistry of lake water and groundwater from the BJD and its periphery, and ruled out the possibility of recharge from recent local rainfall to the lakes and groundwater. In our previous study (Chen et al., 2012), the hydrochemical results were also combined with isotope methods to support our remote recharge hypothesis. In this comment paper, we will add some discussions based on existing studies related to groundwater chemical compositions to further support our hypothesis.

3. Page 2/line 32. Distance between Qilian Mountains and the desert shown on the map (Fig. 3) seems to be smaller than 500 km.

**Response and changes to the manuscript:** 500 km is the approximate distance between the centers of the desert and the Qilian Mountains. This sentence will be revised to be more explicit to avoid misunderstanding.

4. Page 2/lines 35 - 38. The reasoning presented in the last two sentences of page 2 is logically flawed. Incorrect calculation of the averaged isotopic composition of precipitation does not invalidate the meaning and significance of tritium results.

**Response and changes to the manuscript:** Agree. In this paragraph, we meant to approve their results and opinions on the $^{14}$C and tritium dating, which suggests modern rainfall as the source of groundwater. However, when determining the recharge area, their incorrect use of precipitation isotope data may lead to an incorrect conclusion. This paragraph will be rewritten following your comment to improve its logicality.

5. Page 5/lines 113 - 117. Are the surface water bodies mentioned here known to recharge groundwater or do hydrogeological conditions allow for infiltration from them?

**Response and changes to the manuscript:** Based on the isotope evidences available, we hypothesized that the surface water bodies might be a possible source of the groundwater in the BJD. But further studies are needed to further verify this. As to the hydrogeological conditions, large deep fault systems exist in this area and may act as important pathway for the groundwater recharge, as we hypothesized in our previous studies (Chen et al., 2004, 2012). More discussions related to the fault systems and the possible recharge processes will be added in the revised manuscript to address your concerns.

6. Page 5-6/lines 122 - 127. Recharge in Qilian Mountains cannot be a source of detectable tritium in the desert or we have to assume that groundwater flows over hundreds of kilometers in tens of years. As with point 3, tritium data are not well integrated in the discussion.

**Response and changes to the manuscript:** We will add more discussions about the faults and how they may deliver water from the Qilian Mountains to BJD over decades.

Fig. 1. What are standard deviations (due to averaging) of the monthly and annual averages presented here? They should be shown on the plots if significant.

**Response and changes to the manuscript:** The standard deviations of the values will be added in this figure. Also, as suggested by referee #1, information about the seasonality and trends of precipitation isotope will be presented here.

Fig. 2. There is a considerable spread in groundwater isotopic data used to derive EL2 evaporation line, which might lead to a biased identification of the line itself and of its interception with GMWL. These data are pooled results of several studies, do all of them represent locations on the presumed groundwater flow lines between the recharge area and BJD lakes? Perhaps not all of them are representative for derivation of the evaporation line.

**Response and changes to the manuscript:** These data were not chosen to represent locations on the presumed groundwater flow lines between the recharge area and BJD lakes. Not all the data from published studies but only groundwater and lake water samples within the BJD area were chosen for determining the EL2. Therefore the data of groundwater and lake water can be well combined to derive a local EL2 in the BJD and to determine the isotope composition of their origin water. The considerable deviations of groundwater isotopic data may be caused by different mixing rates between the original source and rainwater in lower areas (as we noted in the last paragraph of the paper). We will try to further explain and justify the data selection.

**References**
Chen, J. S., Li, L., Wang, J. Y., Barry, D. A., Sheng, X. F., Zu Gu, W., Zhao, X.

and Chen, L.: Groundwater maintains dune landscape, Nature, 432(7016), 459–460, doi:10.1038/nature03166, 2004.

Chen, J. S., Sun, X. X., Gu, W. Z., Tan, H. B., Rao, W. B., Dong, H. Z., Liu, X. Y. and Su, Z. G.: Isotopic and hydrochemical data to restrict the origin of the groundwater in the Badain Jaran Desert, Northern China, Geochemistry Int., 50(5), 455–465, doi:10.1134/S0016702912030044, 2012.

Yang, X. and Williams, M. A. J.: The ion chemistry of lakes and late Holocene desiccation in the Badain Jaran Desert, Inner Mongolia, China, Catena, 51(1), 45–60, doi:10.1016/S0341-8162(02)00088-7, 2003.

---

## Author Comment (AC2) · 13 Jul 2018

Thanks for your corrections on your original paper as mentioned in the reply. We believe our comment paper and the public discussions can lead to better understanding of the groundwater origin in the Badain Jaran Desert for all interested readers. Although the exact recharge area needs to be further verified, we share the opinion that the groundwater in the desert is mainly recharged by modern precipitation. Based on the analysis presented in our comment paper, we still hold the hypothesis that the modern precipitation on the Qilian Mountains is more likely to be the main source of the groundwater and lake water in the BJD. Whether a preferential recharge from the

extremely scant local winter precipitation (as you suggested) maintaining the abundant groundwater in the BJD could happen needs to be verified.

We have carefully considered your suggestions as well as the comments of two referees and will make revisions accordingly. With regards to your first suggestion, we can add more results of Qilian sourced water (some rivers) as shown in your paper and make related discussions. Your second suggestion will be considered together with the referees' suggestions. We are going to add more descriptions of the hydrogeological conditions and discussions on the recharge and discharge processes of groundwater.

---

## Author Response (AR1)

Dear Editor,

We appreciate greatly the valuable comments from you and the reviewers. We have carefully considered these comments and revised the manuscript accordingly. The details of the revisions and a marked-up version of the revised manuscript are attached to this letter.

Thank you for your consideration of our paper.

Yours sincerely,
Lucheng Zhan
On behalf of the co-authors

**Detailed change list**

**Please refer to the marked-up version of the revised manuscript when checking the revisions.**

**Referee #1**

- A key role in both papers is the data set from the WMO/IAEA on stable isotopes in precipitation – station Zhangye (1986 – 2003, n=86). I recommend, that more information on GNIP station Zhangye and in addition to the mean monthly isotope data set (Fig. 1a) the long-term isotope set is implemented in the work to clarify seasonality and trends of the 17 years data set.
**Changes:** Figure 1 has been modified (Lines 253-258) with addition of more information about the seasonality and trends of precipitation isotope in the study area; and related discussions were added in the text (Lines 52-56).

- Wu et al. were describing their data point as annual average from 'monthly weighted average' values. Therefore I would recommend that the authors include their weighing formula into the text. Were mean monthly values weighed or monthly values to yearly precipitation?
**Changes:** The detailed descriptions of the methods used to calculate the weighted isotopic compositions of precipitation have been added. (Lines 62-71)

- The authors do not comment on earlier an earlier hypotheses, that groundwater might contribute fossil water (Line 30), which potentially was recharged during cooler periods and therefore with more depleted d2H, d18O values. If this would be the case, elevations would not need to be as high as 3914 m a.s.l. (Line 88).
**Changes:** For clarity, more analysis and discussion on the groundwater age and residence time have be added in the revised manuscript to further support our hypothesis. (Lines 35-37, 154-166)

- Line 82: (Figure 2b, c) instead of (Figure 2b&c)
**Changes:** This has been revised as suggested. (Line 98)

- Line 105: Wu et al. 2017 instead of 2016
**Changes:** This mistake has been corrected. (Line 126)

- Line 144: …isotope data of the Zhangye station to determine….
**Changes:** This sentence has been revised. (Line 178)

- Line 217: …monthly precipitation of the GNIP station Zhanye (a). …
**Changes:** This sentence has been revised following the suggestion. (Line 254)

- Line 220: delete: Further details are provided in the text.
**Changes:** This sentence has been deleted. (Line 258)

- Line 222: dD vs. d18O …
**Changes:** This sentence has been changed to "$\delta$D vs. $\delta^{18}$O plot of water related to…" (Line 260)

- Line 223: … (b, c) … instead of (b & c)
**Changes:** It has been revised. (Line 261)

**Referee #2**

1. Precisely describe methods used to calculate both the weighted and unweighted average isotopic compositions of precipitation.
**Changes:** More details about the calculation processes have been added in the text. (Lines 62-71)

2. The two disputed components of groundwater – recent infiltration and water recharging in distant mountain chain - should be easily distinguishable by the concentrations or concentration ratios of dissolved components. Are there any data that could be used to identify their chemical signatures?
**Changes:** We have added some discussions based on existing studies related to groundwater chemical compositions to further support our hypothesis. (Lines 148-153)

3. Page 2/line 32. Distance between Qilian Mountains and the desert shown on the map (Fig. 3) seems to be smaller than 500 km.
**Changes:** This sentence has been revised. (Line 32)

4. Page 2/lines 35 - 38. The reasoning presented in the last two sentences of page 2 is logically flawed. Incorrect calculation of the averaged isotopic composition of precipitation does not invalidate the meaning and significance of tritium results.
**Changes:** This paragraph has been revised following the comment to improve its logicality. (Lines 34-43)

5. Page 5/lines 113 - 117. Are the surface water bodies mentioned here known to recharge groundwater or do hydrogeological conditions allow for infiltration from them?
**Changes:** Large deep fault systems in this area may act as an important pathway for the groundwater recharge. More discussions related to the fault systems and the possible recharge processes have been added in the revised manuscript to address the concerns. (Lines 157-171)

6. Page 5-6/lines 122 - 127. Recharge in Qilian Mountains cannot be a source of detectable tritium in the desert or we have to assume that groundwater flows over hundreds of kilometers in tens of years. As with point 3, tritium data are not well integrated in the discussion.

**Changes:** More discussions about the tritium data and how groundwater transports from the Qilian Mountains to BJD over decades have been added in the text. (Lines 157-166)

Fig. 1. What are standard deviations (due to averaging) of the monthly and annual averages presented here? They should be shown on the plots if significant.

**Changes:** The standard deviations of the values have been added in this figure. (Lines 253-258)

Fig. 2. There is a considerable spread in groundwater isotopic data used to derive EL2 evaporation line, which might lead to a biased identification of the line itself and of its interception with GMWL. These data are pooled results of several studies, do all of them represent locations on the presumed groundwater flow lines between the recharge area and BJD lakes? Perhaps not all of them are representative for derivation of the evaporation line.

**Changes:** Some statements have been added or revised to address the concerns. (Lines 87-89, 173-176)

**X. Wu**

1. This comment make some good theoretic analysis of the altitude effect on the stable isotopes of precipitation, however, I did not see more isotopic data from the Qilian Mountain. Considering they questioned the representability of the samples from Shiyang River (Li et al., 2016), I think more data from the Qilian Mountain would make this comparison more clearly.

**Changes:** More data for Qilian sourced water (some rivers on the northern slope) and related discussions have been added in the revised manuscript. (Lines 119-125, 259)

2. In my opinion, the more specific description of the hydrogeological processes and the evolution of water isotopes is necessary to support the remote Qilian Mountain as the major recharge area.

**Changes:** More descriptions of the hydrogeological conditions and discussions on the recharge and discharge processes of groundwater have been added. (Lines 157-171)

[revised manuscript text omitted]

---

## Author Response (AR2)

Dear Editor,

We have made further corrections as you suggested. We also went through the manuscript again and made some small improvements. The details of the revisions can be found in the marked-up version attached to this letter.

Thank you again for your consideration of our paper.

Yours sincerely,
Lucheng Zhan
On behalf of the co-authors

[revised manuscript text omitted]